# Clinical Impact of Balneotherapy and Therapeutic Exercise in Rheumatic Diseases: A Lexical Analysis and Scoping Review

Lucrezia Tognolo [1] , Daniele Coraci [1], Antonella Fioravanti [2] , Sara Tenti [2], Anna Scanu [1,3] , Giacomo Magro [1], Maria Chiara Maccarone [1] and Stefano Masiero [1,*]

[1] Department of Neuroscience, Physical Medicine and Rehabilitation, University of Padova, Via Giustiniani 2, 35128 Padova, Italy; lucrezia.tognolo@unipd.it (L.T.); daniele.coraci@unipd.it (D.C.); anna.scanu@unipd.it (A.S.); giacomo.magro@studenti.unipd.it (G.M.); mariachiara.maccarone@studenti.unipd.it (M.C.M.)
[2] Rheumatology Unit, Department of Medicine, Surgery and Neuroscience, Azienda Ospedaliera Universitaria Senese, Policlinico Le Scotte, Viale Bracci 1, 53100 Siena, Italy; fioravanti7@virgilio.it (A.F.); sara_tenti@hotmail.it (S.T.)
[3] Rheumatology Unit, Department of Medicine-DIMED, University of Padua, 35128 Padova, Italy
[*] Correspondence: stef.masiero@unipd.it

**Abstract: Objective:** To review the evidence regarding the clinical effect of spa therapy for rheumatic diseases, with particular attention given to association protocols between balneotherapy and rehabilitation interventions, and to support the literature research and studies' selection with lexical analysis. **Methods:** A lexical analysis was performed considering a list of words representing diseases and outcome measures linked to the theme studied in our review. Then, two independent researchers conducted a literature search on PubMed using the string employed for lexical analysis, including Randomized Controlled Trials regarding spa therapy's clinical effects on patients affected by rheumatic diseases published in the last 30 years. After the exclusion of works that did not meet the eligibility criteria, 14 studies were included in the final scoping review. **Results:** Spa therapy has shown a favourable effect on pain, function and quality of life in patients with Osteoarthritis, Fibromyalgia and Rheumatoid Arthritis. Different treatment modalities and types of water have demonstrated beneficial long-term clinical improvement. Furthermore, the association between thermal therapy and rehabilitation treatments has shown better clinical outcomes, probably due to the synergistic effect between the peculiar properties of the thermal waters and the therapeutic exercise program, if conducted in the same context. **Conclusions:** The combination of balneotherapy and rehabilitative interventions seems to be effective in ameliorating several outcomes in patients with rheumatic diseases. However, due to the wide variety of methodologies and interventions employed, these findings need to be further investigated. The lexical analysis should represent an auxiliary support for an extensive evaluation of scientific literature.

**Keywords:** balneotherapy; spa therapy; mud therapy; rheumatic disease; rehabilitation; therapeutic exercise; lexical analysis

## 1. Introduction

Musculoskeletal disorders (MSDs) represent one of the major health problems worldwide for their high prevalence, impact on the quality of life (QoL) and the increased risk of motor disability. The recent analysis of Global Burden of Disease (GBD) data showed that approximately 1.71 billion people globally have MSDs and that these conditions constitute the highest cause in terms of years lived with disability (YLDs) [1,2]. Among them, rheumatic diseases contribute to the largest proportion of musculoskeletal burden, with higher prevalence for five major conditions: Low-Back Pain (LBP), Osteoarthritis (OA), Neck Pain, Rheumatoid Arthritis (RA), and Gout [3].

The management of rheumatic pathologies includes a combination of pharmacological and non-pharmacological interventions to reduce pain, improve functional limitations and ameliorate the QoL. International clinical guidelines generally recommend and highlight the rehabilitation protocols in the complex multimodal treatment of these conditions [4].

Nowadays, one of the most frequently prescribed non-pharmacological treatments in the management of rheumatic diseases is Balneotherapy (BT) [5]. BT is a complementary modality using mineral waters from natural springs, for the preventive, therapeutic, and rehabilitative treatment of various illnesses. It includes a broad spectrum of interventions, such as bathing, applications of peloid, dry peloid therapy, gas bath, drinking, inhalation, irrigations, etc [6].

The BT modalities most frequently used for rheumatic diseases are thermal baths and mud applications. Thermal baths require patient immersion in a tub or pool containing thermal water; a cycle usually ranges from 10 to 20 sessions and can also include hydromassage. Mud therapy consists of applying a layer of mud heated to 45–50°, generally for 20 min.

BT can be combined with different treatment modalities, such as massage, exercise, physical therapy, health education, diet, etc. (Spa therapy) [6]. Furthermore, it is possible to perform a rehabilitative program with hydrokinesitherapy in thermal water with a faster recovery [7–9].

A number of randomized clinical trials, meta-analyses and systematic reviews showed a positive effect of BT and Spa therapy on pain, function and QoL in patients with LBP, OA and Fibromyalgia Syndrome (FMS) [10–12]. However, the available evidence concerning the potential beneficial role of BT in combination with rehabilitation protocols in rheumatic diseases is still sparse.

Additionally, the variety of scientific literature often does not facilitate the review analysis, not only because of the large number of papers but even of the inhomogeneity of the methods and the terms used for defining a disease or a medical intervention. For these reasons, support can come from a wide analysis of the lexicon used in the papers. Recently, we have proposed a method of lexical network based on graph-theory (LENGTH), able to highlight the association among the papers and a list of selected words. The method can also represent the importance of a word or a paper in the network. Hence, it can be used for supporting the paper selection for further review. In a few words, with the LENGTH approach, we performed a two-step analysis. First, we evaluated the most common words conveying meaning about the diseases, the clinical conditions, the outcome measures, the balneotherapy and the rehabilitation and their relationships with the papers. After this first analysis, we used the methods to perform an initial selection of the papers for the review.

We, therefore, aim to review the highest evidence provided by published controlled trials to investigate the clinical effect of spa therapy for rheumatic diseases, with particular attention to association protocols between BT modalities and therapeutic exercise.

## 2. Materials and Methods

### 2.1. Lexical Analysis

A literature search on PubMed using the following string: "(((balneotherapy [MeSH Terms]) OR (peloid therapy [MeSH Terms])) OR (spa therapy [MeSH Terms])) AND (rheumatic disease [MeSH Terms])" was performed. MeSH terms were used to increase the effectiveness of the search. Only Randomized Controlled Trials (RCTs) since 1990 were considered. Then, the proper function of PubMed was used to export the full information (titles and abstracts) of the results. Through the freeware software TXM 0.8.0 the frequency of the words in titles and abstracts was calculated. Hence, we considered a list of words representing the diseases, the clinical conditions, the outcome measures, the balneotherapy and the rehabilitation linked to the theme studied with our review. This list was decided on the basis of the authors' experience and the word frequency as previously calculated. We did not consider words used less than 10 times in the whole list of papers. Finally, we calculated which words were present in each paper (considering the title and abstract).

When a word was contained in a paper, a value of 1 was assigned to the couple word-paper; otherwise, a value of 0 was assigned. With this calculation, a matrix was built, describing the association between the words and the paper. This matrix was imported into the freeware software Gephi 0.9.5. Through graph-theory rules, this can codify the matrix in a graph made of nodes (words or paper) and edges (connections between words and papers) and calculate specific parameters indicating the importance of the nodes [13,14]. In particular, we used the ForceAtlas2 layout to display the graph. Concerning the parameters of node importance, we considered: degree, closeness centrality, betweenness centrality and modularity class. The degree indicates the total absolute frequency of a word or the number of times a paper contains the words in our list. We have additionally calculated the percentage of papers in which a term was contained. The closeness centrality indicates the quality of the connections related to the direct or indirect relationships of the words and the papers. The betweenness centrality is related to the simultaneous presence of two words in a paper or the simultaneous presence of a word in two different papers. For these reasons, the centrality measurements may be used to estimate the potential importance of words and papers in the network. Finally, we considered the modularity class to evaluate the number of communities of the network, which is the number of groups made of strongly related words and papers.

After this first step, we considered the papers containing the terms "exercise" and "rehabilitation". These papers were further evaluated by the authors in order to only select the studies reporting balneotherapy/spa therapy/peloid therapy in combination with usual rehabilitation programs.

*2.2. Scoping Review*

A scoping review was conducted with the aim of searching for evidence of spa therapy plus rehabilitation protocols' clinical effects on patients affected by rheumatic diseases over the last 30 years.

Firstly, the research question was defined by the first author in collaboration with the other authors: "is the combination of spa therapy and therapeutic exercise effective in clinical improvement of patient with rheumatic diseases?". After the research issue had been identified, two independent researchers performed a search on MEDLINE (PubMed) using the string employed for lexical analysis and a comprehensive process of identifying and selecting appropriate studies.

Original research articles that had been published from January 1990 to December 2021 were included. Studies selected for the review needed to have an available abstract and BT as the main treatment under investigation, assess clinical outcomes and be RCTs. Only human subjects with rheumatic diseases were considered. Articles not written in English were excluded.

After the identification of relevant studies, the data were extracted and charted. Papers that did not meet the inclusion criterion were excluded (Figure 1).

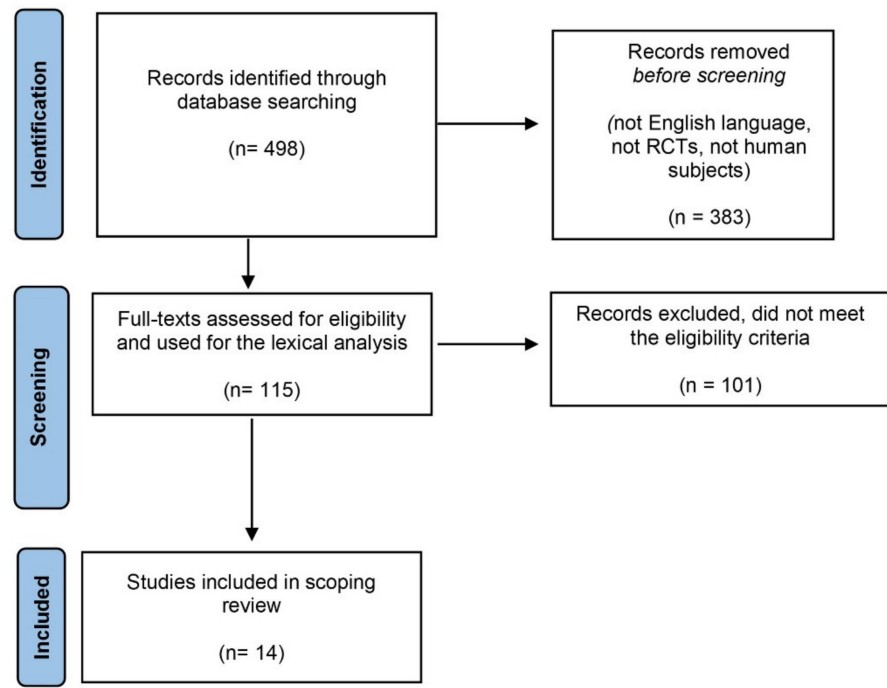

**Figure 1.** PRISMA flow diagram of selection for review.

## 3. Results

The literature search found 115 papers. The selected words presented a wide range of frequency, from 242 of the term "pain" to 16 of the term "disability" (Table 1). Considering the diseases, the graph showed the high frequency of the word "osteoarthritis" followed by "fibromyalgia". "Rheumatoid arthritis" was little represented and mentioned in 19 papers. The words related to the three most mentioned diseases were automatically distributed on the periphery of the graph together with the term "pain". In our model, the less represented and connected outcome measure was "VAS" (Visual Analogue Scale) (Figure 2). Furthermore, considering the centrality measures, "pain" and "osteoarthritis", besides presenting the highest frequencies, additionally presented the highest levels of connections.

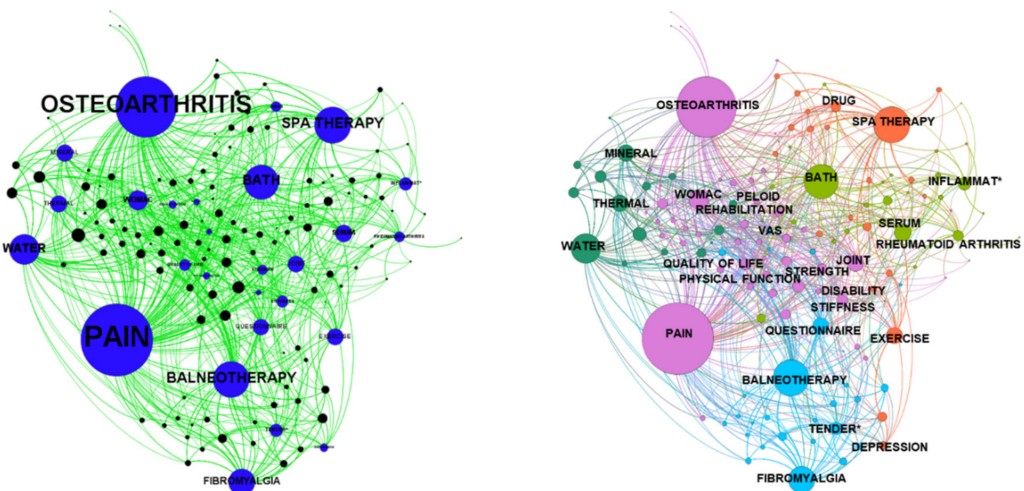

**Figure 2.** Results of the lexical analysis based on graph-theory. On the left side, the blue nodes represent the considered words, the black ones represent the papers. On the right side, graphical representation of the modularity class: each colour indicates the words and the papers more strictly connected in comparison with the whole network.

**Table 1.** The parameters of the considered words calculated by the graph-theory approach. The asterisk indicates the parts of the words with the common root. For example, "INFLAMMAT *" can refer to inflammation or inflammatory.

| Word | Frequency | Percentage of Papers | Closeness Centrality | Betweenness Centrality | Modularity Class |
|---|---|---|---|---|---|
| PAIN | 242 | 71.304% | 0.605 | 2882.023 | 0 |
| OSTEOARTHRITIS | 203 | 55.652% | 0.520 | 1973.552 | 0 |
| SPA THERAPY | 121 | 24.348% | 0.409 | 524.651 | 3 |
| BALNEOTHERAPY | 120 | 33.043% | 0.439 | 614.035 | 2 |
| BATH | 114 | 39.130% | 0.459 | 867.504 | 1 |
| WATER | 99 | 30.435% | 0.431 | 309.482 | 4 |
| FIBROMYALGIA | 87 | 20.000% | 0.382 | 268.424 | 2 |
| WOMAC | 64 | 27.826% | 0.421 | 117.673 | 0 |
| SERUM | 55 | 15.652% | 0.388 | 234.543 | 1 |
| MINERAL | 54 | 22.609% | 0.409 | 138.781 | 4 |
| THERMAL | 54 | 20.000% | 0.399 | 135.333 | 4 |
| EXERCISE | 53 | 14.783% | 0.386 | 170.211 | 3 |
| JOINT | 53 | 27.826% | 0.423 | 414.819 | 0 |
| QUESTIONNAIRE | 52 | 31.304% | 0.434 | 185.071 | 2 |
| INFLAMMAT * | 42 | 16.522% | 0.386 | 260.842 | 1 |
| TENDER * | 41 | 18.261% | 0.393 | 79.258 | 2 |
| STIFFNESS | 37 | 21.739% | 0.406 | 107.399 | 0 |
| QUALITY OF LIFE | 36 | 23.478% | 0.411 | 90.770 | 0 |
| RHEUMATOID ARTHRITIS | 33 | 16.522% | 0.382 | 308.579 | 1 |
| DRUG | 32 | 17.391% | 0.393 | 209.676 | 3 |
| STRENGTH | 31 | 13.913% | 0.384 | 66.528 | 0 |
| DEPRESSION | 25 | 12.174% | 0.370 | 20.494 | 3 |
| REHABILITATION | 25 | 11.304% | 0.378 | 78.571 | 4 |
| PHYSICAL FUNCTION | 22 | 11.304% | 0.374 | 19.066 | 0 |
| PELOID | 19 | 6.087% | 0.353 | 144.595 | 0 |
| VAS | 16 | 12.174% | 0.382 | 31.870 | 2 |
| DISABILITY | 15 | 7.826% | 0.368 | 5.249 | 0 |

Considering the papers containing the terms "exercise" and "rehabilitation", we first selected 28 studies. After the exclusion of works that did not meet the eligibility criteria, 14 studies were included in the final scoping review (Table 2).

**Table 2.** Main characteristics of the included studies.

| First Author (Publication Year) | Pathology | N (M/F) | Intervention | Comparison | Outcomes | Evaluation Times | Main Conclusions |
|---|---|---|---|---|---|---|---|
| *Franke (2007)* | RA | 134 (F) | Radon + $CO_2$ baths | $CO_2$ baths | VAS for everyday life limitation, pain intensity (PI), pain frequency (PF), functional capacity (FC), drug consumption | Baseline, end of the treatment, 4, 8 and 12 months | Favourable changes of pain relief in the radon group until at least 9 months' postintervention compared to baseline |

**Table 2.** *Cont.*

| First Author (Publication Year) | Pathology | N (M/F) | Intervention | Comparison | Outcomes | Evaluation Times | Main Conclusions |
|---|---|---|---|---|---|---|---|
| *Altan (2004)* | FMS | 50 (F) | Pool-based exercise program in a therapeutic pool at 37 °C for 35 min a day three times a week for 12 weeks | Balneotherapy sessions of 35 min three times a week for 12 weeks in the same pool, without any exercise | VAS, morning stiffness, fatigue, sleep disorder parameters of the Hamilton depression scale, tender points, global evaluation of the patient, global evaluation of the physician, FIQ, chair test, BDI | Baseline, 12 and 24 weeks | No significant overall superiority of pool-based exercise over balneotherapy without exercise |
| *Bağdatlı (2015)* | FMS | 70 (F) | Balneotherapy (10 heated pool baths 20 min at 38 °C) and 10 mud-pack applications + 6-h patient education programme on fibromyalgia syndrome | 6-h patient education programme on fibromyalgia syndrome | Patient's Global Assessment, Investigator's Global Assessment, FIQ, pain, fatigue, nonrefreshing sleep, stiffness, anxiety, depression and BDI | Baseline, end of the treatment, 1 month, 3 months | Significant improvements up to the end of the third month in patient's and investigator's global assessment scores, total FIQ score, and pain intensity, fatigue, non-refreshed awaking, stiffness, anxiety and depression subscales of FIQ and BDI as compared to baseline values in balneotherapy group |
| *Zijlstra (2005)* | FMS | 134 (6 M/ 128 F) | Thalassotherapy (seven or eight sessions of: hamam, algotherapy, douche a' affusion, whirlpool, underwater jetstream massage, pool exercise and massage) + Supervised exercise and group education | Treatment as usual | RAND-36, VAS, FIQ, tender points score, 6-min walking test | Baseline, end of the treatment, 3–6–12 months | Significant improvement in symptoms and quality of life lasting for 3 to 6 months in the combination of thalassotherapy, exercise and patient education group |

**Table 2.** *Cont.*

| First Author (Publication Year) | Pathology | N (M/F) | Intervention | Comparison | Outcomes | Evaluation Times | Main Conclusions |
|---|---|---|---|---|---|---|---|
| *Angioni (2019)* | Chronic LBP due to axial OA | 66 (22 M/ 44 F) | Spa therapy including mud packs + Thermal hydrotherapy rehabilitation | Treatment as usual | VAS, SF-36, Roland and Morris Disability Question-naires, Neck Disability Index, Serum proteins concentration | Baseline, 2 weeks, 12 weeks | Significant improvement in VAS pain, Roland Morris disability questionnaire and neck disability index at both time points. A significant increase ($\geq$2.5 fold) after spa treatment of various serum proteins |
| *Fazaa (2014)* | Knee OA | 240 (61 M/ 179 F) | Thermal treatment (underwater showers, massages-jet showers, pool rehabilitation) | Physical rehabilitation treatment + Electrotherapy | VAS, Lequesene AFI score, WOMAC | Baseline, end of the treatment, 12 months | Significant improvement in VAS pain at 12 months in thermal cure group, with superiority of physical treatment with regard to the function component of KOA at 6 months |
| *Fioravanti (2015)* | Bilateral primary knee OA | 103 (29 M/ 74 F) | Mud packs + Mineral baths + Regular care routine (exercise, ac-etaminophen, NSAIDs, SYSADOA, intra-articular hyaluronic acid) | Regular care routine alone (exercise, ac-etaminophen, NSAIDs, SYSADOA, intra-articular hyaluronic acid) | VAS, WOMAC, SF-12, EQ-5D, EuroQol-VAS, drug consumption | Baseline, end of the treatment, 2 weeks, 3–6–9–12 months | Significant superiority in all the assessed parameters at the end of therapy up to 3 months of a cycle of mud-bath therapy in addition to usual treatment over usual treatment alone |

**Table 2.** *Cont.*

| First Author (Publication Year) | Pathology | N (M/F) | Intervention | Comparison | Outcomes | Evaluation Times | Main Conclusions |
|---|---|---|---|---|---|---|---|
| *Forestier (2010)* | Knee OA | 451 (237 M/ 214 F) | Spa therapy (mineral hydrojet, manual massages of the knee and thigh under mineral water, applications of mineral matured mud, supervised general mobilisation in a collective mineral water pool) + Home exercise program | Home exercise program | VAS, WOMAC, SF-36 | Baseline, 1 month, 3 and 6 months | Significant improvement in VAS for pain and WOMAC functional subscale at 6 months' evaluation in the combined treatment group |
| *Forestier (2014)* | Knee and GOA | 214 (89 M/ 25 F) | Spa treatment (general shower, mineral hydro-jet, manual massages, mineral-matured muds) + Supervised mobilisation in a mineral water pool | Three-day wellness package at the spa resort | WOMAC, VAS, SF-36 | Baseline, 3–6–9 months | Better clinically relevant improvement in pain and function in the group treated with spa therapy in combination with a home exercise in comparison with a home exercise programme alone in patients suffering from GOA at 6 months |
| *Gay (2010)* | Knee OA | 123 (22 M/ 101 F) | Spa therapy (mineral hydrojet, thigh massage under mineral water, mineral-matured mud, supervised general mobilisation in a miner pool) + Self-management exercise sessions | Spa therapy alone | IPAQ score, WOMAC, VAS, HAD, ASES, KOFBeQ, EPPA, drug consumption | Baseline and 3 months | Efficacy of a 3-week spa therapy on improvement in physical activity level, with no complementary effect of a self-management exercise program. Significant improvement in the perception of physical activity motivation and barriers subscales, anxiety and depression in the combined treatment group at 3 months follow-up |

**Table 2.** *Cont.*

| First Author (Publication Year) | Pathology | N (M/F) | Intervention | Comparison | Outcomes | Evaluation Times | Main Conclusions |
|---|---|---|---|---|---|---|---|
| *Kasapoğlu Aksoy (2017)* | Hand OA | 63 (NA) | Peloid therapy + Home exercise program | Home exercise program alone | VAS, AUSCAN, HAQ, Hand grip strength, Pinch strength, Radiographs | Baseline, end of the treatment, 1 month | Significant reduction of pain and improvement in functional state, grip strength and quality of life up to 1 month in hand OA patients treated with peloid therapy + home exercise program |
| *Kovács (2016)* | Hip OA | 41 (NA) | Balneotherapy + Home exercise program | Home exercise program alone | WOAMC, EuroQol-5D | Baseline, end of the treatment, 12 weeks | Statistically significant improvement at 12 weeks in favour of balneotherapy group in WOMAC pain, stiffness, function a total score |
| *Rat (2020)* | Knee OA | 283 (93 M/ 190 F) | Spa therapy + 3 week Rehabilitation Program aimed to improve joint mobility, strength and proprioception and develop appropriate behavior and attitudes | Standard Spa therapy (mineral hydrojet sessions, manual massages under mineral water, applications of mineral-matured and supervised general mobilization in a collective mineral water pool) | VAS, WOMAC, PASS, SF-36, OAKHQOL, pROM | Baseline, 3 weeks, 6 weeks, 3 and 6 months | The non-inferiority of spa-rehab versus standard spa therapy for the main outcome criteria at 6 months could not be demonstrated. Spa-rehab therapy was not inferior to standard spa therapy for the MCII criteria at 3 months or the PASS at 3 and 6 months. Therefore, spa-rehab therapy may be an acceptable alternative to standard spa therapy for patients with symptomatic knee OA |

**Table 2.** *Cont.*

| First Author (Publication Year) | Pathology | N (M/F) | Intervention | Comparison | Outcomes | Evaluation Times | Main Conclusions |
|---|---|---|---|---|---|---|---|
| *Yurtkuran (2006)* | Knee OA | 56 (3 M/ 53 F) | Balneotherapy + Home based exercise program | Heated tap water therapy + Home based exercise program | VAS, Fifty-foot walking time, Active knee flexion, TS, Quadriceps muscle strength, WOMAC, NHP | Baseline, end of the treatment, 12 weeks | Significant improvement in all the variables except TS and muscle strengh at 2 and 12 weeks in the heated tap water group. The improvement observed in the BT group was superior to tap water group for pain VAS, TS, and psychological and emotional variables at the 2nd and 12th week |

RA: Rheumatoid Arthritis. FMS: Fibromyalgia Syndrome. $CO_2$: Carbon Dioxide. VAS: Visual Analogue Scale. FIQ: Fibromyalgia Impact Questionnaire. BDI: Beck Depression Inventory. SF-36: Short Form health survey of 36 items. NSAIDs: SYSADOA: Symptomatic Slow Acting Drugs for OA. SF-12: Short Form health survey of 12 items. EQ-5D: European Quality-of-Life Questionnaire-5 Dimensions. EuroQol: Euro Quality of life. GOA: Generalized Osteoarthritis. IPAQ: International Physical Activity Questionnaire. HAD: Hospital Anxiety and Depression scale. ASES: Arthritis Self-Efficacy Scale. KOFBeQ: Knee Osteoarthritis Fears and Beliefs Questionnaire. EPPA: Evaluation of the Perception of Physical Activity. AUSCAN: Australian/Canadian Hand Osteoarthritis Index. HAQ: Health Assessment Questionnaire. PASS: Patient Acceptable Symptom State. OAKHQOL: Osteoarthritis Knee and Hip Quality Of Life questionnaire. pROM: passive Range of Motion. MCII: Minimal Clinically Important Improvement. TS: Tenderness Score. NHP: Nottingham Health Profile.

*3.1. Osteoarthritis*

A total of 11 studies evaluating the effect of a combined therapeutic protocol in patients affected by OA were included in the final analysis.

Rat and colleagues recently proposed a "spa-rehab" protocol, combining usual spa therapy and rehabilitation care in patients with knee OA [15]. They observed that the combination of spa and rehabilitation interventions was not inferior to standard spa therapy alone at 3-month follow-up in improving pain (VAS score) and functionality (WOMAC—Western Ontario and McMaster University—score,). However, more patients seemed to lose the clinically significant effect at 6 months, probably due to the short effect of rehabilitation.

The association of BT and exercise therapy achieved a more sustained improvement of joint function, enhancement in QoL, and decrease in pain than exercise therapy alone [16,17]. Particularly, in patients with hip OA, it appears that the combination of balneotherapy and home exercise therapy has a more sustained effect (12 weeks) than exercise therapy alone, in improving pain, joint stiffness and function [16].

Forestier et al. first evaluated the effect of spa therapy combined with a home-based exercise protocol in subjects with both knee and general OA on symptoms related to general OA [17]. Although the improvements in the patient acceptable symptom state or quality of life were not sufficient to reach a significant improvement in the medium (3 months) and long-term follow-up (6 months), the authors reported a positive trend in the group who underwent the combined protocol.

Fioravanti and colleagues conducted a long-term (1 year) follow-up study on patients with bilateral knee OA. They compared the addition of a cycle of bath-mud therapy to the usual care (including physiotherapy) to usual treatments alone [18]. Only patients

who underwent the combined treatments showed a progressive improvement in VAS and WOMAC physical function scores up to 3 months. However, at 1-year follow-up, there was a slight not significant increase in WOMAC physical function in both groups, probably due to the natural history of OA.

Patients treated with BT combined with a home-based exercise program compared to the same treatment in tap water demonstrated to have significantly superior effects on pain, tenderness score and the Nottingham Health Profile [19].

A 3-week BT program together with home exercises and usual pharmacological treatments demonstrated to offer benefits on pain and function persisting after 6 months [10].

A 12-month improvement of painful symptoms of knee OA has been demonstrated through underwater shower, hydromassage, pool rehabilitation and peloid therapy [20].

In patients with chronic back pain secondary to axial osteoarthritis, a cycle of 12 mud packs applied on the body surface followed by a bicarbonate-alkaline mineral water bath or a thermal hydrotherapy rehabilitation applied 12 times was found to have a clear clinical benefit. Even the combination of the two interventions was demonstrated to be effective [21].

In hand OA, peloid therapy for a total of 10 sessions and home exercise programs have been demonstrated to be effective in improving hand grip strength, hand functions, and QoL [22].

A recent multicentric trial evaluated the synergistic effect of a self-management exercise program associated with spa therapy in patients affected by knee OA [23]. The authors observed that a self-management exercise program had no significant complementary effect on the improvement in physical activity level in participants with KOA compared with spa therapy. However, spa therapy treatment, including mineral hydro jet sessions, massages under mineral water, applications of mineral-matured mud and supervised general mobilisations in a collective mineral pool, may help to increase the level of physical activity of patients with knee OA.

Lexical Analysis

The lexical evaluation confirmed the remarkable scientific impact of the disease. The term was highly frequent, just less common than the term "pain". The word was mentioned 203 times in 55.65% of the collected papers. This disease was the most assessed in our paper sample. Furthermore, it presented high values of centrality, indicating a strong relationship with the other papers and words. Additionally, the disease showed high connections with "WOMAC" and several outcome measures and clinical signs related to the main fields of rehabilitation. (Figure 2). Furthermore, as visible in the graph on the right, osteoarthritis presented the major connections with pain and the terms related to physical functions (e.g., strength, disability).

### 3.2. Fibromyalgia

A total of three studies evaluating the combination of balneotherapy or spa therapy and therapeutic exercise were included.

Two weeks of BT applications combined with patient education and a home-based exercise programme showed more beneficial effects compared to patient education alone [24]. The BT treatment consisted of both heated pool baths and mud-pack applications, while the home-based program was individualized and included stretching, relaxing, and posture exercises.

The study by Zijlstra et al. aimed to assess the effects of a group programme of thalassotherapy, exercise, and self-management education in patients with FMS, both in terms of fibromyalgia-related symptoms and of health-related quality of life. The treatment programme included a Turkish bath, hot packs with algae, massage while lying under a shower, whirlpool, underwater jet-stream massage, and supervised group pool exercise. The authors observed that a combination of thalassotherapy, exercise and patient education may temporarily (up to 3 months) improve FMS symptoms and health-related QoL [25].

The study by Altan et al. compared the effect of pool-based exercise with balneotherapy alone in a population of 50 female patients who were diagnosed with FMS [26]. In patients with FMS, pool-based exercise did not show a significant superiority over BT without exercise, even if it seemed to have a longer-lasting effect on some of the symptoms. Particularly, the balneotherapy group was superior at the end of the first month regarding pain intensity, patient's global assessment, fatigue, sleep quality, stiffness, anxiety, depression, and Fibromyalgia Impact Questionnaire (FIQ) scores.

Lexical Analysis

The lexical assessment revealed the term presence in 20.00% of the papers, making fibromyalgia the second most common disease assessed in the selected papers. Although with this frequency, we found a lower level of the centrality of this term in comparison with "rheumatoid arthritis" (see below). "Fibromyalgia" apparently showed a lower level of connections with the other elements. This was particularly visible for the betweenness centrality and should be associated with the relatively low presence of this term in different papers. Indeed, the word was common but with more repetitions in the same documents, as the percentage of papers shows. Interestingly, fibromyalgia was associated with "tender*" (Figure 2). The lexical analysis revealed that this condition was associated with the specific term "balneotherapy".

### 3.3. Rheumatoid Arthritis

Only one RCT regarding patients with RA has been included in the scoping review.

Franke et al. conducted a study aimed to investigate the effects of radon (plus $CO_2$) baths on RA in contrast to artificial $CO_2$ baths combined with a complex rehabilitative program [27]. The authors evaluated the effects on self-assessed limitation in everyday life, pain intensity and frequency, morning stiffness, functional capacity and drug consumption up to one year after the end of the treatment.

Radon baths in addition to a multimodal rehabilitative treatment including exercises, physiotherapy, occupational therapy and galvanic baths have demonstrated beneficial long-term effects with long-lasting improvements in pain intensity, as well as reduced consumption of corticosteroids and NSAIDs and/or analgesics.

Lexical Analysis

The lexical evaluation revealed rheumatoid arthritis was present in 16.52% of the papers but cited 33 times. The term showed a relatively high level of connections with the other nodes of the graph. The term was strictly associated with "serum" and "inflammat*" (Figure 2).

### 4. Discussion

The use of lexical analysis allows extensive evaluation and speculation about the words used in the papers. For example, considering fibromyalgia, the strict association with the word "tender" may be related to the typical evaluation of fibromyalgia, which is based on the identification of specific tender points. Regarding rheumatoid arthritis, the strict association with the inflammation may be explained by the common impairment of the joints characterized by the peculiar inflammation. The association with the term "serum" is interesting, because it may be related to the high interest in the use of serum outcome measures for the evaluation of patients with rheumatoid arthritis.

Besides these considerations, with the LENGTH approach, the words can be classified not only for their frequency but also for their importance in the whole literature about the topic. It can be promising support for a scoping review. In fact, the lexical analysis displays the formal characteristic of the current scientific production and can allow discussions about the ways in which the information is spread in the literature. Furthermore, it may help the selection of the papers, focusing attention on the works containing specific terms.

Our analysis showed an important concern regarding the use of proper terminology. Indeed, by applying the lexical analysis method, only a few works, compared to the multitude of the initial results, included the researched keywords.

Therefore, only 14 studies have been included in the scoping review. Moreover, due to the heterogeneity of the studies, regarding waters/mud content, treatment periods, protocols, and potential associations with other treatments, it is not possible to define with certainty the superiority of a combined BT and exercise program intervention in rheumatic diseases. However, the results showed some beneficial effects of BT treatments that could positively modify impairments and symptoms related to rheumatic diseases.

These positive effects may probably be the result of a combination of mechanical (hydrostatic force), thermal (mineral water or mud temperature), and chemical (mineral water or mud composition) effects. Immersion in thermal water through increased buoyancy can lead to an unloading effect, which is protective in degenerated and/or inflamed joints. Hydrostatic pressure can have a positive effect on muscle tone, joint function, and pain intensity. Temperature, stimulating cortisol, adrenocorticotropic hormone, prolactin and β-endorphin release, may have an important analgesic action and an immunosuppressive effect [28].

Studies carried out in cellular models of acute joint inflammation demonstrated that treatment with hydrogen sulphide donors reduced oxidant-induced mitochondrial dysfunction [29], and the production of pro-inflammatory mediators [30,31]. Similar immunomodulatory properties were observed also after incubation with NaHS or thermal water of human OA chondrocytes or fibroblast-like synoviocytes [32–36].

In vivo experiments with mud therapy and bath treatments showed a significant reduction in TNF-α and IL-1 serum levels and decreased leukocyte migration into the articular cavity in rat models of OA [37,38]. In vitro and in vivo studies also showed a positive action of mineral waters on the oxidant/antioxidant system, especially after incubation in sulphurous mineral water [28].

An in vivo study performed on C57BL/6N mice that spontaneously develop OA has demonstrated that mud-therapy potentiated the effect of chondroitin sulphate by reducing the serum level of nitric oxide (NO) [39]. Furthermore, mud therapy can have a protective effect on OA articular cartilage through an increase in chondrocyte number and collagen amount [40].

The positive effects described in OA patients after mud-bath therapy can also be linked to a decrease in adiponectin and resistin levels, involved in the development and progression of the disease [8].

In vivo studies also confirmed that the application of mud and treatment with mineral water may have a positive effect on pain perception in terms of attenuation of mechanical hyperalgesia and enhanced levels of endogenous molecules modulating neuropathic pain [41–43].

Finally, many findings highlight the therapeutic advantages related to the positive social atmosphere of the spa setting on chronic diseases, reducing mental stress and improving resilience and QoL [44].

Surprisingly, the research failed to find any studies addressing ankylosing spondylitis. Although it is a highly disabling rheumatic condition, the research string selected to investigate the clinical impact of BT in rheumatic diseases did not find any RCTs related to this condition. Further studies may include a broader terminology (e.g., MSDs) to investigate BT's role also in ankylosing spondylitis.

A comprehensive approach including traditional BT modalities and rehabilitation interventions in the spa environment could be effective in treating patients with rheumatic diseases, and consequent acute or chronic disability, exploiting synergies between BT properties and exercise or physical therapy [7,45,46].

This scoping review presents some limitations. First, the use of specific limited keywords and the application of the lexical analysis, although selecting only studies using proper terminology, provided only a few studies. Moreover, the selected articles

were heterogeneous in the type of thermal therapy intervention proposed (spa therapy, balneotherapy, mud packs, etc.) and therapeutic exercise protocols (supervised vs home-based, individualized vs group exercise, etc.). Excluding non-controlled clinical trials may be another limitation, since pool-based exercise and rehabilitation protocols, in general, may have been described in manuscripts not included in our search.

## 5. Conclusions

The combination of different BT modalities to various therapeutic exercise protocols could represent an effective multimodal strategy in ameliorating several outcomes in patients affected by FMS, OA and RA. However, due to the wide variety of methodologies and interventions employed, these findings need to be further investigated. Moreover, due to the heterogeneity of the studies, regarding water/mud content, treatment periods, protocols, and potential associations with other treatments, it is not possible to define with certainty the superiority of a combined BT and exercise program intervention in rheumatic diseases. However, the results showed some beneficial effects of BT treatments that could positively modify impairments and symptoms related to rheumatic diseases.

Further studies will help to define the best interventions for the treatment of each rheumatic disease. Finally, the lexical analysis should represent an auxiliary support for an extensive evaluation of the scientific literature. It is a method able to show and summarize the relationships between the papers and the words used to define a topic. It is an alternative way that does not substitute the usual revision process, but it analyses the issue from a formal point of view. In particular, it provides information about the most frequent words written in the papers and how these words are usually associated. The information may be useful to better understand the strong and weak points of the literature and how specific contexts are or could be conveyed. Finally, the methods show if similar meanings are expressed with different words and which words are more informative.

**Author Contributions:** Conceptualization, S.M. and A.F.; methodology, L.T. and D.C.; investigation, L.T., M.C.M., S.T. and G.M.; data curation, L.T., D.C. and M.C.M.; writing—original draft preparation, L.T., D.C. and M.C.M.; writing—review and editing, S.T., A.S. and G.M.; supervision, S.M. and A.F. All authors have read and agreed to the published version of the manuscript.

**Funding:** This research received no external funding.

**Institutional Review Board Statement:** Not applicabile.

**Informed Consent Statement:** Not applicabile.

**Conflicts of Interest:** The authors declare no conflict of interest.

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
