# Peer review of "Clinical Impact of Balneotherapy and Therapeutic Exercise in Rheumatic Diseases: A Lexical Analysis and Scoping Review"

_applsci, doi:10.3390/app12157379_

Round 1

Reviewer 1 Report

Tognolo et al reviewed the BT therapy from their understanding in this work, however, from a scientific aspect, the current work is hard for readers to get the key points and fail to fulfill the requirement of a review paper, please find the following details:

1, By carefully reading the paper, from the introduction, there is no clear clue why this paper is important, why the authors could write the review for this field;

2, Regarding the methods, and discussion, the review felt hard to understand the clue of this work, and can not find the current problems and solutions in this work, but a file of abstracts from other works.

3, For the conclusion‘ Balneotherapy seems to be effective in ameliorating several clinical outcomes in patients with rheumatic diseases, also combined to rehabilitative interventions’ , how come ‘seems to be’ appear in conclusion? 

Based on the above questions, the reviewer can not find potential contribution of this paper to this field, thus can not provide a more positive suggestion.

Author Response

Tognolo et al reviewed the BT therapy from their understanding in this work, however, from a scientific aspect, the current work is hard for readers to get the key points and fail to fulfill the requirement of a review paper, please find the following details:

1, By carefully reading the paper, from the introduction, there is no clear clue why this paper is important, why the authors could write the review for this field;

Thank You for the useful observation. We modified the Title (by including “Therapeutic Exercise”), we have expanded the Background to explain our purpose and why this study can be useful and we specified the aim of the study (please see Introduction, lines 75 to 90).

2, Regarding the methods, and discussion, the review felt hard to understand the clue of this work, and can not find the current problems and solutions in this work, but a file of abstracts from other works.

Thank You for the suggestion. The paper was widely improved. In particular, we have used the described method of lexical analysis for two main steps. First, we used it to describe the overview the current literature. Second, we have used it to support the paper selection. We focused on the papers about the balneotherapy and the usual rehabilitation programs. For this selection, the papers containing the terms “exercise” and “rehabilitation” were considered. Then, among them, we only selected the papers which actually assessed the potential added values of the association of balneotherapy and the usual rehabilitation. The method has been better described.

3, For the conclusion‘ Balneotherapy seems to be effective in ameliorating several clinical outcomes in patients with rheumatic diseases, also combined to rehabilitative interventions’ , how come ‘seems to be’ appear in conclusion? 

Thank You for the observation. We modified this section on the basis of the thorough revisions of the paper.

Reviewer 2 Report

This manuscript studies the Clinical Impact of Balneotherapy in Rheumatic Diseases, using a Lexical Analysis and Scoping Review

The presence of errors in the bibliographical references make the manuscript in its current state unsuitable for publication.

Major comments:

Missing references [79 to 99]. The manuscript cannot be revised from lines 342 to 377.

Figure 2A is a Table. A Table 1 must be included in MDPI format (I suggest adding a column in that Table with the % of papers in which the term appears)

Lines 393 to 544. References are incomplete, correct all to MDPI format

If the manuscript does not undergo a major revision, in my opinion it is not suitable for publication.

Minor comments:

Line 44: where it says (2); should put [2]

Line 127: where it says (Fig. 2A); you should put Table 1

Line 132: where it says Fig. 2B; you must put Fig. 2A

Lines 135-136: Figure 2A should be Table 1 (MDPI format)

Lines 137-141: Figures B and C, become: Figures A and B. Better explain caption

Line 191: possible citation error [39] (corresponds to Fibromyalgia) (lines 467-468) Should it be changed?

Line 235: where it says Fig. 2; you must put Table 1 and Fig. 2

Line 287: where it says Fig. 2; you must put Table 1

Line 306: remove citation [37] (corresponds to OA)

Line 319: where it says Fig. 2; you must put Table 1 and Fig. 2

Author Response

Major comments:

Missing references [79 to 99]. The manuscript cannot be revised from lines 342 to 377.

Thank You for the observation. We fixed the errors and provided the missing references. Since we thoroughly revised the Manuscript, total number of references has been modified.

Figure 2A is a Table. A Table 1 must be included in MDPI format (I suggest adding a column in that Table with the % of papers in which the term appears)

Thank You for the suggestion. We have revised, building a proper table. The calculation of the percentage of papers where the terms were used was also added.

Lines 393 to 544. References are incomplete, correct all to MDPI format

We have revised all the references that have been formatted in MDPI style.

If the manuscript does not undergo a major revision, in my opinion it is not suitable for publication.

Minor comments:

Line 44: where it says (2); should put [2]

Thank You for the observation, we fixed the error.

Line 127: where it says (Fig. 2A); you should put Table 1

We modified Fig.2A into Tab. 1.

Line 132: where it says Fig. 2B; you must put Fig. 2A

Figure 2 ha been modified. We replaced Fig.2B with Fig.2

Lines 135-136: Figure 2A should be Table 1 (MDPI format)

Lines 137-141: Figures B and C, become: Figures A and B. Better explain caption

Line 191: possible citation error [39] (corresponds to Fibromyalgia) (lines 467-468) Should it be changed?

Line 235: where it says Fig. 2; you must put Table 1 and Fig. 2

Line 287: where it says Fig. 2; you must put Table 1

Line 306: remove citation [37] (corresponds to OA)

Line 319: where it says Fig. 2; you must put Table 1 and Fig. 2

We thank the Reviewer for the careful analysis of the paper and the relevant suggestions. Since we widely revised and modified the main, most of the previous lines and references do not correspond to the actual. About the figures and table, Fig. 1 and 2 and Tab. 1 has been modified and we consequently fixed the errors highlighted by the Reviewer. The incorrected and not relevant citations have been modified and the citation errors been fixed.

Reviewer 3 Report

In this review of the evidence regarding the "clinical impact" of balneotherapy in musculoskeletal diseases, an original and interesting approach of "lexical analysis" was used, but the review of the evidence itself is a classical narrative short review of the papers selected in a way not directly involving the lexical analysis. Regarding the selection process (Prisma graph) of the reviewed papers, a "research question" is mentioned but not clearly described, and the "eligibility criteria" that allowed the elimination of one fourth of the papers are not listed. This makes the value of the review and the conclusions questionable.

As stated by the authors, "the lexical analysis displays the formal characteristic of the current scientific production and can allow discussion about the ways in which the information is spread in the literature." This reviewer cannot see how this could "confirm the remarkable scientific impact of the disease" (osteoarthritis). A more detailed description of the "lexical analysis" method and a thorough discussion of the added value of the information, compared to the results published with this method in other fields of medicine and therapeutics would have been more relevant.

Author Response

In this review of the evidence regarding the "clinical impact" of balneotherapy in musculoskeletal diseases, an original and interesting approach of "lexical analysis" was used, but the review of the evidence itself is a classical narrative short review of the papers selected in a way not directly involving the lexical analysis.

Thank You. We have better integrate the “classical” review process and the lexical analysis. We have used the described method of lexical analysis for two main steps. First, we used it to describe the overview the current literature. Second, we have used it to support the paper selection.

Regarding the selection process (Prisma graph) of the reviewed papers, a "research question" is mentioned but not clearly described, and the "eligibility criteria" that allowed the elimination of one fourth of the papers are not listed. This makes the value of the review and the conclusions questionable.

 We thank the Reviewer for the relevant and useful observation. We widely revised and modified the paper (find in the main text highlighted in yellow the changes). We therefore better define the research question and the selection criteria of the studies, also based on the lexical analysis.

As stated by the authors, "the lexical analysis displays the formal characteristic of the current scientific production and can allow discussion about the ways in which the information is spread in the literature." This reviewer cannot see how this could "confirm the remarkable scientific impact of the disease" (osteoarthritis). A more detailed description of the "lexical analysis" method and a thorough discussion of the added value of the information, compared to the results published with this method in other fields of medicine and therapeutics would have been more relevant.

Thank You for Your relevant comment. We have improved the entire manuscript considering, for the lexical analysis, the diseases, the clinical conditions, the outcome measures and the terms related to balneotherapy and rehabilitation. We have better described the method and discussed the results.

Reviewer 4 Report

This work announces to seek to achieve two objectives, on the one hand, to show the interest of the lexical analysis and, on the other hand, to evaluate the contribution of rehabilitation techniques to the programs of balneotherapy.

The first point is speculative but not devoid of interest.

The second objective has strong medical and medico-economic implications.

In the current state of the drafting, it is not clear that these two objectives have been achieved.

Thus, the interest of lexical analysis remains, in my opinion, insufficiently argued. It is not clear what this analysis really brings to this work. This is an aspect that must imperatively be reworked in an eventually revised version.

Regarding the purely medical dimension, the work seems on the one hand too diffluent, considering too many pathologies for too few studies, on the other hand exposing clinical conclusions presented as robust certainties whereas they do not exceed the stage of possibilities to be confirmed by more conclusive data.

It is obvious that the choice to analyze only the texts of the abstracts and titles of the articles is not appropriate to the approach.

We know that the use of the term "spa therapy" usually implies the association with pure hydrothermal treatments (baths, showers, mud, ...) of physical techniques made of massage, exercise, ...

The authors eliminated clinical trials that did not mention the word exercise in the title or the abstract. Thus, by way of example, escaped the most important RCT analysis published on the thermal management of fibromyalgia (2021), a study in which patients had exercise practices in their program, …

The results and the discussion do not really address the concrete interest and contribution of lexical analysis in this work.

The discussion does not sufficiently question the demonstrative weakness of the clinical benefit analysis. The limitations of the work are exposed in a few lines at the end of the discussion.

It is difficult to understand the long exposition (lines 309-350), presented in the discussion, on the mechanisms of action. It does not seem related to the aims of the article and what is more, is not based on concrete elements provided by the study. These elements, on the other hand, would constitute an excellent rationale for an article intended to present the results of a study carried out with a view to demonstrating a therapeutic mechanism. The motivations for such an exposition in this paper remains for me unclear.

The best would be that the authors should resume work, limited to a single pathology, but extending the lexical analysis of the entire text to all studies that include the term "Spa therapy" in their title, and comparing them to studies that do not include it.

Author Response

This work announces to seek to achieve two objectives, on the one hand, to show the interest of the lexical analysis and, on the other hand, to evaluate the contribution of rehabilitation techniques to the programs of balneotherapy. The first point is speculative but not devoid of interest.

Thank You, we have implemented considering the following comments of Your attentive revision (green highlighted in the text).

The second objective has strong medical and medico-economic implications. In the current state of the drafting, it is not clear that these two objectives have been achieved. Thus, the interest of lexical analysis remains, in my opinion, insufficiently argued. It is not clear what this analysis really brings to this work. This is an aspect that must imperatively be reworked in an eventually revised version.

The lexycal analysis evaluates the issue from a formal point of view. In particular, it provides information about the most frequent words written in the papers about a theme and how these words are usually asociated. Therefore, it provides useful information to better understand the strong and weak points of the literature and how specific contexts are, or could be, conveyed. Moreover, the method shows if similar significants are expressed with different words and which words are more informative.

Regarding the purely medical dimension, the work seems on the one hand too diffluent, considering too many pathologies for too few studies, on the other hand exposing clinical conclusions presented as robust certainties whereas they do not exceed the stage of possibilities to be confirmed by more conclusive data.

It is obvious that the choice to analyze only the texts of the abstracts and titles of the articles is not appropriate to the approach.

We agree with the Reviewer. However, the use of titles and abstracts, beside representing the first step for paper selection in a review, is always possible, being titles and abstracts present in the major search engines (like PubMed).

We know that the use of the term "spa therapy" usually implies the association with pure hydrothermal treatments (baths, showers, mud, ...) of physical techniques made of massage, exercise, ...

The authors eliminated clinical trials that did not mention the word exercise in the title or the abstract. Thus, by way of example, escaped the most important RCT analysis published on the thermal management of fibromyalgia (2021), a study in which patients had exercise practices in their program, …

The results and the discussion do not really address the concrete interest and contribution of lexical analysis in this work.

We agree with the Reviewer. However the present research aims to aanalyze the RCTs concerning the combination of therapeutic exercise and balneotherapy by selecting only the papers that use a proper terminology. The limited number of incuded works is due to the frequent use of an improper terminology both in the titles and in the abtracts. Indeed, as reported by the SPIRIT checklist for interventional trials, the title should have the following features: “descriptive title identifying the study design, population, interventions and, if applicable, trial acronym”. Therefore, it is clear why a paper on thermal management combined to exercise that does not mention the word “exercise” in its title has not been selected by our research.

The discussion does not sufficiently question the demonstrative weakness of the clinical benefit analysis. The limitations of the work are exposed in a few lines at the end of the discussion.

It is difficult to understand the long exposition (lines 309-350), presented in the discussion, on the mechanisms of action. It does not seem related to the aims of the article and what is more, is not based on concrete elements provided by the study. These elements, on the other hand, would constitute an excellent rationale for an article intended to present the results of a study carried out with a view to demonstrating a therapeutic mechanism. The motivations for such an exposition in this paper remains for me unclear.

The best would be that the authors should resume work, limited to a single pathology, but extending the lexical analysis of the entire text to all studies that include the term "Spa therapy" in their title, and comparing them to studies that do not include it.

We thank the Reviewer for the observations.

According to Gutenbrunner et al (Int J Biometeorol. 2010 Sep;54(5):495-507), we use the world Balneotherapy in the title, because we finally selected  only the studies including thermal baths (combined or not, to other spa therapies, such as muds, etc.). Indeed, Balneotherapy represents a central treatment modality of a spa therapy regimen, and the long exposition (lines 309-350) has the aim to better explain the rationale of the benefits of using the thermal water in the rehabilitation program of these pathologies. Therefore, in these lines, we have briefly exposed the possible mechanisms of action on different biologicals processes due to the mineral and thermal  properties of these waters.

Round 2

Reviewer 1 Report

Thanks for the revision of the paper, however, as mentioned previously, the reviewer does not think this work could be a scientific review paper, but a popular science article. Therefore, the reviewer would insist on the previous decision. 

Author Response

The lexycal analysis represents a validate method able to show and summarize the relationship between the papers and the words used to define a topic. The method is an alternative  way that does not substitute the usual revision process, but it analyzes the issue from a formal point of view. In particular, it provides information about the most frequent words written in the papers about a theme and how these words are usually asociated. The information may be useful to better understand the strong and weak points of the literature and how specific contexts are, or could be, conveyed. Finally, the method shows if similar significants are expressed with different wors and which words are more informative.

Moreover, the choice of study design (a scoping review) is due to the fact that scientific literature is lacking of good quality RCTs investigating the advantages of the combination of balneotherapy in thermal water and physical exercise in MSK diseases. However, even worse is the use of an improper terminology, as demonstrated by the poor number of works by using a MesH terminology and by our lexical analysis.

Reviewer 2 Report

Please, check error in Figure 1. (n=493? or n=383?)

Author Response

We thank the Reviewer for the observation. The error has been fixed, the correct number of identified papers is 498: of these, 383 have been removed before screening procedure.